# Carrier-Free Cellular Transport of CRISPR/Cas9 Ribonucleoprotein for Genome Editing by Cold Atmospheric Plasma

**DOI:** 10.3390/biology10101038

**Published:** 2021-10-13

**Authors:** Haodong Cui, Min Jiang, Wenhua Zhou, Ming Gao, Rui He, Yifan Huang, Paul K. Chu, Xue-Feng Yu

**Affiliations:** 1Materials Interfaces Center, Shenzhen Institute of Advanced Technology, Chinese Academy of Sciences, Shenzhen 518055, China; hd.cui@siat.ac.cn (H.C.); min.jiang@siat.ac.cn (M.J.); ming.gao@siat.ac.cn (M.G.); yf.huang@siat.ac.cn (Y.H.); 2University of Chinese Academy of Sciences, Beijing 100049, China; 3Department of Physics, City University of Hong Kong, Tat Chee Avenue, Kowloon, Hong Kong, China; paul.chu@cityu.edu.hk; 4Department of Materials Science and Engineering, City University of Hong Kong, Tat Chee Avenue, Kowloon, Hong Kong, China; 5Department of Biomedical Engineering, City University of Hong Kong, Tat Chee Avenue, Kowloon, Hong Kong, China

**Keywords:** cold atmospheric plasma, CRISPR/Cas9, carrier-free transport, genome editing

## Abstract

**Simple Summary:**

CRISPR/Cas9 system as a potential gene editing platform has been widely applied in biological engineering and disease therapies. To achieve precise gene targeting, active CRISPR/Cas9 components must be efficiently transported to targeted cells. As a simple and effective strategy, Cold Atmospheric Plasma (CAP) treatment has been demonstrated for the transmembrane delivery of various exogenous materials. In comparison with carrier-dependent delivery methods, this carrier-free platform provides a promising alternative to circumvent the obstacles of biosafety and complicated preparation processes. In this work, a CAP-based CRISPR/Cas9 carrier-free delivery platform has been established and corresponding mechanism related to efficient transportation has been explored. Briefly, the efficient production of bioactive species in culture media after CAP treatment alters cell membrane potential and permeability, which facilitates cytosolic delivery of active CRISPR/Cas9 components via passive diffusion and ATP-dependent endocytosis pathways, resulting in efficient genome editing and gene silencing. This carrier-free strategy using CAP-based transportation may also be extended to other active biomolecules in drug delivery and gene therapy.

**Abstract:**

A carrier-free CRISPR/Cas9 ribonucleoprotein delivery strategy for genome editing mediated by a cold atmospheric plasma (CAP) is described. The CAP is promising in many biomedical applications due to efficient production of bioactive ionized species. The MCF-7 cancer cells after CAP exposure exhibit increased extracellular reactive oxygen and nitrogen species (RONS) and altered membrane potential and permeability. Hence, transmembrane transport of Ca^2+^ into the cells increases and accelerates ATP hydrolysis, resulting in enhanced ATP-dependent endocytosis. Afterwards, the increased Ca^2+^ and ATP contents promote the release of cargo into cytoplasm due to the enhanced endosomal escape. The increased membrane permeability also facilitates passive diffusion of foreign species across the membrane into the cytosol. After CAP exposure, the MCF-7 cells incubated with Cas9 ribonucleoprotein (Cas9-sgRNA complex, Cas9sg) with a size of about 15 nm show 88.9% uptake efficiency and 65.9% nuclear import efficiency via passive diffusion and ATP-dependent endocytosis pathways. The efficient transportation of active Cas9sg after the CAP treatment leads to 21.7% and 30.2% indel efficiencies in HEK293T and MCF-7 cells, respectively. This CAP-mediated transportation process provides a simple and robust alternative for the delivery of active CRISPR/Cas9 ribonucleoprotein. Additionally, the technique can be extended to other macro-biomolecules and nanomaterials to cater to different biomedical applications.

## 1. Introduction

Cas9 (CRISPR-associate protein 9) is an RNA-guided endonuclease protein associated with the CRISPR (clustered regularly interspaced short palindromic repeats) system, which is involved in bacteria adaptive immunity against invading nucleic acids [1]. Owing to the simplicity and robustness, the CRISPR/Cas9 platform has been widely applied to transcription regulation [2,3], gene editing [4,5] and in situ labeling of genomic loci [6,7]. Various approaches of CRISPR/Cas9 have been developed to deliver DNA fragments encoding Cas9 ribonucleoprotein (Cas9-sgRNA complex, Cas9sg) via microinjection [8,9], electroporation [10,11], or viral vectors [12,13]. However, long-term exposure of foreign DNA fragments and associated overexpression of functional Cas9sg may consequently increase the risk of insertional mutagenesis [14] as well as off-target effects [15]. To overcome these obstacles, cytosolic delivery of Cas9sg is a promising alternative with therapeutic potentials [16,17,18,19,20,21,22]. Although the carrier-dependent delivery can effectively transport Cas9sg into cells, the biosafety of carriers and complicated preparation process to assemble the necessary delivery platform are obstacles stifling biomedical applications. Therefore, new strategies of carrier-free delivery of Cas9 ribonucleoprotein with excellent bioactivity are highly desirable.

The cold atmospheric plasma (CAP) operated at ambient temperature has attracted great interest in biomedical fields such as drug delivery and cancer therapy [23,24]. As one of common configurations, a plasma jet containing a high-voltage electrode and carrier gas is used to generate CAP for biomedical applications. When it is applied to cells, it produces multiple effects that were caused by the electric field, ultraviolet (UV) radiation, charges, and reactive oxygen and nitrogen species (RONS) [25]. Among these effects, the high concentration of actives species in the culture media caused by CAP exposure is believed to be a major factor modulating the membrane permeability [26], effective killing of cancerous cells [27,28,29], as well as elevated transmembrane delivery of DNA [30,31,32] and nanoparticles [33,34]. The carrier-free delivery by electroporation has been reported with high editing efficiency [10], but the complicated devices and significant cell death limited it to in vitro applications.

Herein, a carrier-free Cas9 ribonucleoprotein delivery approach based on CAP for genome editing is described. After exposure to CAP under the proper conditions, the MCF-7 cancer cells incubated with Cas9sg show 88.9% cytosolic delivery efficiency and 65.9% nuclear import efficiency. The corresponding mechanism is discussed and RONS-induced passive diffusion and ATP-dependent endocytosis pathways are clarified. The active Cas9sg is released from endosome into cytoplasm due to the presence of intracellular Ca^2+^ and ATP contents, and then imported to the nuclei directed by nuclear localization sequence (NLS) within Cas9 protein resulting in gene editing efficiencies of 21.7% in HEK293T normal cells and 30.2% in MCF-7 cancer cells, respectively. The highly effective CAP transportation may be extended to other active biomolecules in drug delivery and gene therapy.

## 2. Materials and Methods

Chemicals: Calcein-AM, propidium iodide (PI), imidazole, 3-(4,5-dimethyl-2-thia- zolyl)-2,5-diphenyl-2-H-tetrazolium bromide (MTT) and 2′,7′-dichlorodihydrofluorescein diacetate (DCFDA) were obtained from Sigma-Aldrich (Munich, Germany). Isopropyl β-D-1-thiogalactop- yranoside (IPTG), 4% paraformaldehyde (4% PFA), RIPA lysis buffer, Hank’s Balanced Salt Solution (HBSS), kanamycin, ATP colorimetric assay kit, and 4′,6-diamidino-2-pheny- lindole (DAPI) were bought from Beyotime Biotechnology (Shanghai, China). Alexa Fluor 488 NHS Ester (Succinimidyl Ester) and Lipofectamine CRISPRMAX Cas9 Transfection Reagent were bought from Thermo-Fisher Scientific (Waltham, MA, USA). UltraPure™ 1 M Tris-HCI Buffer (pH 7.5) was purchased from Invitrogen (Carlsbad, CA, USA). MβCD, sodium chloride, tiron, sucrose, and amiloride were obtained from Aladdin Industrial Corporation (Shanghai, China). Dimethyl sulfoxide (DMSO) and ethanol were purchased from Sinopharm Chemical Reagent Co., Ltd. (Shanghai, China). FDA, Rh123, mannitol, Reactive Oxygen Species Assay Kit, sodium pyruvate (SP) and NAC were bought from Yeasen Corporation (Shanghai, China). The enzymes were obtained from NEB (Ipswich, MA, USA). Phosphate-buffered saline (PBS, pH 7.4), FBS, DMEM, penicillin-streptomycin, and trypsin-EDTA were purchased from Gibco Life Technologies (Basel, Switzerland). All the other chemicals used in this study were analytical reagent grade and used without purification. Ultrapure water (18.25 MΩ.cm, 25 °C) was used to prepare all the solutions.

*Plasma jet system:* The CAP in this study was generated by a home-built plasma jet system containing a high-voltage power supply (CTP-2000K, Coronalab, Nanjing, China), gas flow controller (ThreeTek LZB-3WBF, Suzhou, China), oscilloscope (Tektronix TBS1102, Beaverton, OR, USA), and plasma jet device. A powered copper electrode (diameter of 2 mm) was enclosed in a quartz dielectric tube (inner/outer diameter 2/4 mm), which was driven by an AC high-voltage supply at a discharge voltage of 10 kV and frequency of 10 kHz. A grounded electrode was wrapped around the nozzle of the outer dielectric tube with a cooper sheet. An argon gas (high purity, 99.999%) flow of 2 SLM was employed to generate plasma jet.

*Cas9 protein expression and labeling with Alexa-488:* The pET28a/Cas9-Cys expression vector (Addgene, Cambridge, MA, USA, Plasmid 53261) was used for the expression of Cas9 protein using *Escherichia coli* strain BL21 (DE3) (Novegan, Shanghai, China) according to a standard protocol. In brief, the BL21 (DE3) cells transformed with pET28a/Cas9-Cys plasmid were cultured overnight at 30 °C in LB media supplemented with IPTG (0.5 mmol L^−1^) and Kanamycin (50 μg mL^−1^). The cells were centrifuged at 8000 rpm for 15 min (4 °C) and lysed by probe sonication on ice (37% duty, operating for 3 s with the interval of 2 s for a total of 30 min) in the lysis buffer (20 mmol L^−1^ Tris-HCl at pH 8.0, 300 mmol L^−1^ NaCl, 20 mmol L^−1^ imidazole, 1× protease inhibitor cocktail). The solution obtained with centrifuge (10,000 rpm, 15 min, 4 °C) was transferred to chromatography columns (Bio-Red) and incubated with the Ni-NTA agarose resin (Qiagen, Hilden, Germany) for 1 h (4 °C). After two rounds of washing with lysis buffer, Cas9 protein was eluted with an elution buffer (20 mmol L^−1^ Tris-HCl at pH 8.0, 300 mmol L^−1^ NaCl, 250 mmol L^−1^ imidazole, 1× protease inhibitor cocktail), and subjected to dialysis overnight at 4 °C (dialysis buffer: 20 mmol L^−1^ Tris-HCl at pH 8.0, 150 mmol L^−1^ KCl, 25 mmol L^−1^ TCEP, 0.1% *w*/*w* Tween 20). The purity of the eluted protein was analyzed using 6% SDS-PAGE gel and quantitated by the Bradford assays (Bio-Red, Hercules, CA, USA). With regard to protein labeling, the purified Cas9 protein was transferred to PBS by ultrafiltration, and then labeled with Alexa Fluor 488 NHS Ester (Succinimidyl Ester, Alexa-488) based on the manufacturer’s instructions.

*sgRNA design and synthesis:* The single guided RNA (sgRNA) was in vitro transcribed from the T7 promoter driven sgRNA expression vector pDR274 (Addgene, Cambridge, MA, USA, Plasmid 42250). Two sgRNAs were designed to target a sequence within the enhanced green fluorescent protein (EGFP) gene and human genome. Briefly, a pair of oligonucleotides containing customized 20 nt targeting sequence was annealed and inserted into the BsaI-digested backbone downstream of the T7 promotor and then transformed into *E. coli* DH5α. The constructed plasmids were extracted and subjected to the DraI digestion. The digested products (290 bp) were gel extracted and underwent in vitro transcription using the MEGAscript™ T7 Transcription Kit (Invitrogen, Carlsbad, CA, USA) according to the manual. The transcribed sgRNA was ethanol precipitated and quantified by spectrometry. The sequences of annealed oligonucleotides are listed in the following:

EGFP-For: TAGGGCGAGGGCGATGCCACCTA

EGFP-Rev: AAACTAGGTGGCATCGCCCTCGC

Chrom9-T1-For: TAGGCCCAGACTGAGCACGTGA

Chrom9-T1-Rev: AAACTCACGTGCTCAGTCTGGG

(EGFP-For/Rev and Chrom9-T1-For/Rev pairs were used to generate sgRNAs for gene disruption at EGFP and Target 1 sites.)

*Cas9sg binding and cleavage activities:* To assemble the Cas9-sgRNA complex (Cas9sg), the purified Cas9 protein (100 nmol L^−1^) and sgRNA (200 nmol L^−1^) (Targeting a sequence within human Chrom9-T1: GGCCCAGACTGAGCACGTGATGG) were incubated in 1× NEBuffer 2 for 30 min at 37 °C. Afterwards, the Cy5-labeled DNA bearing target sequence was added to give a concentration of 150 nM, and the mixture was incubated at 37 °C for another 30 min. The binding and cleavage activities analyzed with 6% PAGE gel were explored with 10 μL of reaction containing 2 μL 6× loading buffer and 10× SDS loading buffer, respectively.

*Cas9sg assembly for CAP-mediated cell transport:* The Cas9sg was assembled by incubating 1.6 μmol L^−1^ Cas9 protein (or Alexa-488 labeled Cas9) and 3.2 μmol L^−1^ sgRNA in 1× NEBuffer 2 for 30 min at 37 °C. 5 μL of the assembled Cas9-sgRNA complexes (Cas9sg and Cas9sg-488) were suspended in 95 μL of DMEM to give a final concentration of 80 nmol L^−1^.

*Characterization:* The morphology of Cas9sg was examined by atomic force microscopy (AFM) (Bruker MultiMode 8, Bremen, Germany) and the zeta potentials and dynamic size were determined by DLS with Zetasizer (Malvern Zetasizer Nano ZS90, Malvern, UK). The optical properties of the Alexa-488 labeled Cas9 ribonucleoprotein (Cas9sg-488) were evaluated by fluorescence spectroscopy (HITACHI F-4600, HITACHI, Tokyo, Japan) (excitation by 488 nm) and absorbance measured by UV−vis spectrophotometer (HITACHI U-3900, HITACHI, Tokyo, Japan). The optical measurements were carried out at room temperature.

*Cell culture:* The human cell lines HEK293T (embryonic kidney), MCF-7 (breast cancer), A549 (lung cancer), and EGFP expressing A549 (A549/EGFP) were purchased from Cellbank (Shanghai, China). The cells were cultured in DMEM or RPMI-1640 (A549 and A549/EGFP) media supplemented with 10% fetal bovine serum (FBS), 0.1% penicillin-streptomycin at 37 °C in a humidified 5% CO_2_ atmosphere.

*Cellular viability:* MTT assays were utilized to assess the cellular viability. Briefly, the cells were seeded on 24 well plates (5 × 10^4^ per well) and cultured in 500 μL 10% FBS DMEM overnight. The distance between the plasma jet and media level was 25 mm. After CAP exposure for different time periods, the cells were treated with Cas9sg and cultured for 24 h. The culture media were replaced with 400 μL of fresh 10% FBS DMEM containing 0.5 mg mL^−1^ MTT. After incubation for 4 h at 37 °C, the media were removed and replaced with 400 μL DMSO, followed by gentle shaking for 15 min at room temperature. The DMSO mixture was transferred to a new 96-well plate and the absorbance at 490 nm was measured by an ELISA reader (Multiskan GO, Thermo Scientific, Waltham, MA, USA). The cellular viability was also evaluated by calcein-AM/PI staining. Briefly, the cells were seeded on 24 well plates (5 × 10^4^ per well) and cultured overnight, followed by the CAP treatment and continuous culturing for 24 h. Afterwards, the cells were rinsed with PBS and incubated with 2 μg mL^−1^ calcein-AM and 3 μg mL^−1^ PI (dispersing in PBS). The viable or dead cells were observed by fluorescence inverted microscopy (OlympusIX71, Olympus, Tokyo, Japan) with 485 nm (calcein-AM) and 535 nm excitation (PI), respectively.

*CAP-mediated cellular transport:* The CAP-mediated transport efficiency was evaluated by in vitro fluorescence imaging. In brief, the MCF-7 cells (1 × 10^5^ per well) were seeded on 24-well plates and cultured overnight. After CAP exposure for different time, 50 µL of the assembled Cas9sg-488 were added dropwise to each well (8 nmol L^−1^, total in 500 µL of the media) and cultured for different time points at 37 °C. After two rounds of washing with PBS, the cells were examined with 488 nm excitation for Alexa-488 by fluorescence inverted microscopy. To assess the nuclear import efficiency, the MCF-7 cells (5 × 10^4^ per well) were seeded on 24-well plates with climbing glasses and cultured overnight. After the CAP treatment for 80 s, 50 µL of the assembled Cas9sg-488 were dropped on each well and cultured for different time intervals. After rinsing with PBS and fixing with 4% PFA, the cells were stained with nucleus dyes (DAPI, 5 μg mL^−1^) for 10 min at room temperature. After two rounds of washing with PBS, the cells were imaged under 405 nm excitation for DAPI and 488 nm for Alexa-488 by confocal laser scanning microscopy (Leica TCS SP5-II, Leica Microsystems, Wetzlar, Germany). The nuclear transport efficiency was counted and calculated by ImageJ and the nuclear import efficiency was evaluated with the Leica LAS AF Lite software from about 100 cells in each sample (*n* = 3).

*Lipofectamine CRISPRMAX Cas9 transfection:* The Lipofectamine CRISPRMAX transfection reagent was employed to transfect Cas9sg into cells according to the provided transfection protocol. In brief, 0.5 µg of Cas9 and 0.125 µg of sgRNA were mixed in 25 µL of the Opti-MEM media, followed by addition of 1 µL of Cas9 Plus reagent and incubation for 5 min at room temperature (Tube1). Afterwards, 1.5 µL of Lipofectamine CRISPRMAX reagent were added to 25 µL of the Opti-MEM media and incubated for 5 min at room temperature (Tube2). The Cas9sg Plus mixture from Tube1 was mixed with the Lipofectamine CRISPRMAX solution from Tube2 and incubated for 10 min at room temperature. Subsequently, 50 µL of the prepared Cas9sg complex was dropped on each well for efficient transfection (24-well plate, 5 × 10^4^ per well, 500 µL DMEM in total, 6.25 nmol L^−1^ in each well). Similarly, after culturing for different time points, the cells were stained with DAPI and imaged by confocal laser scanning microscopy (Leica TCS SP5-II, Leica Microsystems, Wetzlar, Germany).

*Flow cytometry of CAP-mediated cellular transport:* The cellular uptake was quantitatively determined by flow cytometry (BD FACSCANTO II, BD Biosciences, San Jose, CA, USA). The CAP-treated MCF-7 cells were incubated with Cas9sg-488 at a concentration of 8 nmol L^−1^ for 24 h at 37 °C. After rinsing with PBS, the cells were digested and re-suspended in 200 μL for PBS for flow cytometry (the channel of Alexa-488). The data were analyzed with the FlowJo Analysis software.

*Mechanism of CAP-mediated cellular transport:* The MCF-7 cells (5 × 10^4^ per well, 24-well plates) were pre-treated with MβCD, sucrose, and amiloride or cultured at a low temperature for 0.5 h, followed by CAP exposure for 80 s. The cells without CAP treatment were settled as control groups. Cas9sg-488 was added to each well to yield a final concentration of 8 nmol L^−1^ and cultured 2 h at 37 °C. After rinsing with PBS, the cells were imaged under 488 nm excitation for Alexa-488 by fluorescence inverted microscopy. To investigate the importance of RONS, prior to CAP exposure, the RONS scavenger N-acetyl-L-cysteine (NAC) (1 μmol L^−1^) was added as the RONS-inhibition groups. To quantify the fluorescence intensity, the cells were collected and lysed with RIPA for 0.5 h on ice.

*Extracellular RONS levels:* The MCF-7 cells (5 × 10^4^ per well) were seeded on 24-well plates and cultured overnight, followed by replacement of 500 μL DMEM media supplemented with 10% FBS. 500 μL of 10% FBS DMEM in each well served as the group without cells. After the CAP treatment, 100 μL of the media with or without cells were collected to detect the extracellular RONS levels with DCFDA probe. The DCFDA powder was dissolved in DMSO to make a stock solution (10 mmol L^−1^) and stored at −20 °C. The MCF-7 cells were treated with the DCFDA probe (1 μmol L^−1^ in 100 μL of the media) and incubated for 30 min at 37 °C. The fluorescence intensity was measured on a fluorescence spectrophotometer (485 nm excitation). In the RONS-inhibition experiment, NAC (1 μmol L^−1^) was added to the DMEM media prior to CAP treatment, and the cells were pre-treated with sodium pyruvate (SP) (10 mmol L^−1^), mannitol (50 mmol L^−1^), and Trion (20 μmol L^−1^) to inhibit generation of H_2_O_2_ and NO, OH^−^ and O^2−^ from CAP, respectively. The effects of UV radiation alone were monitored by addition of a quartz plate to block active species.

*Intracellular RONS generation:* The MCF-7 cells (5 × 10^4^ per well) were seeded on 24-well plates and treated with CAP for 80 s. A quartz plate was employed to evaluate the effect of UV radiation alone. After incubation for 2 h, the cells were rinsed with PBS and treated with the DCFH-DA (Reactive Oxygen Species Assay Kit, 10 μmol L^−1^ in 300 μL PBS) and DAF-FM DA (Nitric Oxidative Synthase Assay Kit, 5 μmol L^−1^ in 300 μL buffer) probes to measure H_2_O_2_/NO generation on a fluorescence inverted microscope (485 nm excitation). The fluorescence intensity was further determined by a fluorescence spectrophotometer.

*Immunofluorescence:* The immunofluorescence experiments were processed to clarify the cellular uptake and cytosolic delivery of Cas9sg. Briefly, the MCF-7 cells (5 × 10^4^ per well) were seeded on 24-well plates with climbing glasses and cultured overnight. After CAP exposure for 80 s, 50 µL of Cas9sg were added dropwise to each well (8 nmol L^−1^, total in 500 µL of the media) and cultured for different time points at 37 °C. After two rounds of washing with ice-cold PBS, the cells were fixed by 100% methanol (chilled at −20 °C) for 5 min and subsequently washed three times with PBS. Then, the cells on climbing glass were blocked 1 h with the blocking buffer (containing 0.1% Triton X-100) and simultaneously incubated with the primary antibodies: Caveolae, Clathrin, Rab34, Rab5, LAMP1 (Rabbit), and Cas9 (Mouse). After that, the Alexa Fluor 647 (Anti-Rabbit, red) and 488 (Anti-Mouse, green) labeled secondary antibodies were used to detect the primary antibodies. Finally, the cells on climbing glasses were stained with DAPI and imaged by confocal laser scanning microscopy.

*Determination of the membrane potential and permeability:* The membrane potential and permeability were determined by fluorescence imaging with Rh123 and FDA, respectively. In brief, the MCF-7 cells (5 × 10^4^ per well) were seeded on 24-well plates and cultured overnight, followed by CAP treatment for 80 s and culturing at 37 °C. The Rh123 and FDA powder were dissolved in DMSO as a stock solution (10 mg mL^−1^ for Rh123 and 25 mg mL^−1^ for FDA; stored at −20 °C). After rinsing with PBS, the cells were treated with Rh123 (10 μg mL^−1^ in 300 μL PBS, 30 min incubation) and FDA (0.5 mg mL^−1^ in 300 μL PBS, 5 min incubation) probes at 37 °C, respectively. Subsequently, the MCF-7 cells were imaged by fluorescence inverted microscopy (505 nm excitation for Rh123 and 490 nm excitation for FDA). The fluorescence intensity was quantitatively determined on a fluorescence spectrophotometer.

*Intracellular Ca^2+^ levels:* The Fluo 3-AM probe was utilized to measure the intracellular Ca^2+^ levels after the CAP treatment. Briefly, the MCF-7 cells (5 × 10^4^ per well) were seeded on 24-well plates and cultured overnight, followed by CAP exposure for 80 s and incubating for another 2 h at 37 °C. The Fluo 3-AM powder was dissolved in DMSO as a stock solution (1 mmol L^−1^, stored at −20 °C). The cells were rinsed with HBSS PBS and stained with Fluo 3-AM probe (1 μmol L^−1^ in 300 μL HBSS) for 20 min at 37 °C. Afterwards, 1 mL of HBSS supplemented with 1% FBS was added to each well and incubated for another 40 min, followed by rinsing with HBSS and imaging by fluorescence inverted microscopy (490 nm excitation). The fluorescence intensity was evaluated by a fluorescence spectrophotometer.

*Measurement of the intracellular ATP contents:* The intracellular ATP contents were measured by the ATP colorimetric assay kit according to the manufacturer’s instruction. In brief, the MCF-7 cells (5 × 10^4^ per well) were seeded on 24-well plates and cultured overnight. After the CAP treatment for different time durations, the cells were rinsed and treated with 50 μL of the lysis buffer and centrifuged at 12,000 rpm for 5 min at 4 °C, and 20 μL of the supernatant were transferred to a new 96-well plate and 100 μL of the working solution were added and shaken for 2 s. The ATP content was measured by the Luminometer (GEMINI EM, Molecular Devices, Sunnyvale, CA, USA). The standard curves of ATP concentration were acquired from ATP standard samples and the intracellular ATP contents were calculated based on the standard curve. To further clarify the association of ATP and endocytosis, the cells pre-treated with endocytosis inhibitors and at a low temperature of 4 °C.

*EGFP gene disruption assays:* The A549/EGFP cells (constantly expressing EGFP) were utilized to assess gene disruption of Cas9sg after the CAP treatment. Briefly, the A549/EGFP cells were seeded on 24-well plates (5 × 10^4^ per well) and cultured overnight. After the CAP treatment for different durations, the cells were incubated with 8 nmol L^−1^ Cas9sg for 24 h (targeting the coding region of EGFP, EGFP site sequence: GGCGAGGGCGATGCCACCTACGG). The media were replaced with a fresh and cultured for four more days. EGFP disruption was evaluated by fluorescence inverted microscopy (488 nm excitation of Alexa-488) and the cells were collected and analyzed by flow cytometry.

*Indel analysis:* The 293T and MCF-7 cells (5 × 10^4^ per well) were seeded on 24-well plates and cultured overnight. After the CAP treatment, the cells were incubated with Cas9sg (8 nmol L^−1^) (targeting the human Chrom9-T1 sequence: GGCCCAGACTGAGCACGTGATGG) for 24 h. The media were replaced with a fresh and cultured for two more days. The cells were digested for genomic DNA extraction by the Tissue DNA Kit (Omega Bio-tek, Cambridge, MA, USA), followed by the amplification of specific loci containing double-strand breaks with genomic PCR (primer pairs, SA-Chrom9-T1-For: CTTGTAGCTACGCCTGTGATGGGCT, SA-Chrom9-T1-Rev: TGAGGCTGGCCCCTTCCAGG). The PCR products were gel extracted, re-annealed, and digested by T7 endonuclease-I according to the standard protocol. The digestion products were visualized by a 6% native PAGE gel and the genome editing efficiency was calculated with ImageJ (NIH, Bethesda, MA, USA).

*Statistical analysis:* The results were presented as means ± standard deviation and at least three groups of parallel tests were taken in the experiments (*n* = 3). The significance of difference was analyzed by the one-way analysis of variance (one-way ANOVA) based on Tukey’s post-test. In the statistical evaluation, *p* < 0.05 indicated significant (*), *p* < 0.01 indicated highly significant (**) and *p* < 0.001 indicated very highly significant (***).

## 3. Results and Discussion

### 3.1. CAP Devices and Cas9sg Characterization

As shown in Figure 1a, the CAP is generated by a home-built plasma jet system containing a high-voltage power supply, oscilloscope, and argon carrier gas (2 SLM -standard liters per minute regulated by a gas flow controller). Briefly, a powered copper electrode (diameter of 2 mm) was enclosed in a quartz dielectric tube (inner/outer diameter 2/4 mm), which was driven by an AC high-voltage supply at a discharge voltage of 10 kV and frequency of 10 kHz (Figure 1b,c). A grounded electrode was wrapped around the nozzle of the outer dielectric tube with a cooper sheet. An argon gas (high purity, 99.999%) flow of 2 SLM was employed to generate plasma jet. The plasma emission spectra in the wavelength ranging from 300 to 900 nm were measured (Appendix A). Excited atoms of the feed argon gas were mainly between 696.5 and 912.3 nm. Peaks from N_2_ in surrounding ambient air along the plasma jet were observed between 330 and 380 nm. OH radical at 309 nm were also detected, likely due to H_2_O dissociation. The CAP is applied directly to cells for cellular transport of Cas9sg to induce specific gene editing. The Cas9 protein is expressed with *E. coli.* BL21 (DE3) and further incorporated with sgRNA to assemble the active Cas9sg. Binding and cleavage of Cas9sg are verified in vitro (Appendix A). The active Cas9sg show an average size of 15.08 ± 1.35 nm according to atomic force microscopy (AFM) (Figure 1d). As determined by dynamic light scattering (DLS), the average hydrodynamic size and surface potential are 14.15 ± 1.55 nm (Figure 1e) and −14.6 ± 0.67 mV (Appendix A), respectively.

### 3.2. CAP-Mediated Cellular Transport of Cas9sg

To explore CAP-mediated cellular transport of Cas9 ribonucleoprotein, the Alexa-488 labeled Cas9 protein (Cas9-488) is employed in the assembly of the Cas9 ribonucleoprotein (Cas9sg-488). The optical properties of the assembled Cas9sg-488 are evaluated based on the absorbance and fluorescence spectra (Appendix A). CAP exposure for 20, 40, and 80 s all produce low cytotoxicity in cells with or without Cas9sg after incubation for 24 h according to the MTT assays (Appendix A). Parallel calcein-AM/PI staining experiments also reveal robust green fluorescence from calcein-AM stained MCF-7 cells after the CAP treatment confirming that the CAP treatment up to 80 s does not induce cytotoxicity (Appendix A). To assess the CAP-mediated transport efficiency, the MCF-7 cells are treated with the CAP in the presence of media for different time periods, and then incubated with 8 nmol L^−1^ Cas9sg-488 for 24 h. As shown in Figure 2a, intracellular green fluorescence of Cas9sg-488 from MCF-7 cells increases gradually with CAP exposure time and incubation time, and is significantly higher than that observed from the untreated group (Appendix A). Flow cytometry reveals 88.9% uptake efficiency of Cas9sg-488 in the group subjected to 80 s exposure compared to only 22.1% observed from the untreated group (Figure 2b). To evaluate the nuclear import efficiency, the MCF-7 cells are exposed to CAP for 80 s incubated with 8 nmol L^−1^ Cas9sg-488 for different time intervals and subjected to confocal fluorescence imaging. As shown in Figure 2c and Appendix A, Cas9sg-488 with green fluorescence is gradually transferred from cytosol to the nuclear with increasing incubation time. The nuclear import efficiency increases from 12.8% after incubation for 4 h to 33.8% and 65.9% after 12 and 24 h, respectively (Figure 2d). Moreover, the CAP transport approach reveals the similar uptake and nuclear import efficiencies of Cas9sg-488 via Lipofectamine-mediated transfection (Appendix A). Without CAP exposure, the cellular uptake and nuclear import of Cas9sg-488 are both extremely low, thus, confirming that CAP can significantly improve the cellular uptake and nuclear targeting of Cas9sg.

### 3.3. RONS-Mediated Endocytosis-Dependent and -Independent Uptake of Cas9sg

To clarify the potential mechanism of CAP-mediated cellular transport of Cas9sg, the endocytosis pathway is investigated. Before CAP exposure, the MCF-7 cells are pre-treated either at a low temperature to block all ATP-dependent endocytosis or with different endocytosis inhibitors to block the specific endocytosis pathways. Briefly, the pre-treated cells were cultured at 37 °C for 2 h in the presence of 8 nmol L^−1^ Cas9sg-488, and imaged by fluorescence microscopy, lysed and measured by fluorescence spectroscopy to quantitively determine cellular uptake of Cas9sg. To investigate the biological functions of RONS towards different endocytosis pathways, all the cell groups are pre-treated with RONS scavenger N-acetyl-L-cysteine (NAC) prior to CAP exposure. As shown in Figure 3a, the cells pre-treated at a low temperature and incubated without NAC only exhibit 60% fluorescence reduction, suggesting that CAP-enhanced cellular uptake of Cas9sg-488 is partially mediated by ATP-dependent endocytosis. The endocytosis inhibitors, amiloride, methyl-β-cyclodextrin (MβCD), and sucrose, are utilized to verify the macropinocytosis, caveolae-mediated endocytosis, and clathrin-mediated endocytosis, respectively. Less fluorescence of Cas9sg-488 is observed from cells treated with MβCD and sucrose, suggesting that caveolae-mediated and clathrin-mediated endocytosis constitutes the main endocytosis pathways. Moreover, the cells pre-treated with NAC show decreased fluorescence signals from Cas9sg-488, suggesting that RONS generated from CAP indeed contribute to the enhanced endocytic transport of Cas9sg. As shown in Figure 3b, after CAP exposure, the fluorescence intensity decreases to 35.8% after incubation at a low temperature and 11.6% after pre-treatment with NAC. Furthermore, the cells without CAP treatment show lower fluorescence signals from Cas9sg-488 and less changes in fluorescence signals after being pre-treated with inhibitors (Appendix A) as compared with cells treated with CAP, suggesting that the endocytosis of Cas9sg-488 is very weak in cells without CAP exposures. These results suggest that endocytosis is the major cellular uptake pathway. Moreover, about 24.2% uptake of Cas9sg-488 is enhanced by RONS from CAP exposure through the endocytosis-independent pathway.

To obtain direct evidence showing the importance of RONS in modulating endocytosis-dependent and -independent uptake of Cas9sg, the RONS levels in the media are determined according to the fluorescence intensity of the DCFDA probe, which gives fluorescence signals when interacting with RONS. As shown in Appendix A, strong fluorescence from DCFDA is observed from the media exposed to CAP. On the other hand, the DCFDA intensity in the media decreases significantly decreased after pre-treatment with the RONS scavenger NAC and other scavengers, suggesting that RONS in the media are mainly generated from CAP and increase gradually with CAP-treatment time (Appendix A). In this process, FBS (fetal bovine serum) in the media is essential to CAP-induced RONS generation (Appendix A). Moreover, the RONS levels in the media with cells are lower than those observed from the media without cells, suggesting that RONS are partly consumed by cells to induce the corresponding cellular response. The intracellular ROS and RNS levels evaluated with the DCFH-DA and DAF-FM DA fluorescent probes show robust green fluorescence from cells after the CAP treatment (Figure 3c; Appendix A). Less fluorescence from DCFH-DA and DCF-FM DA treated cells is observed with UV radiation alone and pre-treatment with NAC (Appendix A). These results suggest that RONS is the main factor for the enhanced cellular uptake of Cas9sg based on endocytosis-dependent and -independent pathways.

### 3.4. Intracellular Trafficking of Cas9sg

To further clarify the internalization pathways and subsequent cytosolic release of Cas9sg mediated by CAP, the immunofluorescence experiments are performed to trafficking the co-localization of Cas9sg and three internalization-associated markers (Caveolae, Clathrin, and Rab34) [35]. As shown in Figure 4a, some co-localization of Cas9sg with Caveolae (caveolae-mediated endocytosis) and Clathrin (clathrin-mediated endocytosis) are observed, while hardly any co-localization is visible between Cas9sg and Rab34 (macropinocytosis), suggesting that Cas9sg are delivered into cells by CAP based on caveolae-mediated and clathrin-mediated endocytosis pathways. Meanwhile, after CAP-exposed cells pre-treating with three endocytosis inhibitors (amiloride, MβCD, and sucrose) and at a low temperature of 4 °C, the co-localization of Cas9 with three endocytosis markers are vanished (Figure 4b,c), further confirming that caveolae-mediated and clathrin-mediated endocytosis are the main uptake pathways. However, some Cas9sg are also distributed into cytoplasm without any co-localization, which indicated that Cas9sg enters cells by direct transmembrane via passive diffusion. To confirm the effects of RONS in transportation, cells are pre-treated with NAC and show few Cas9sg signals in cytoplasm (Appendix A), which revealed that RONS generated from CAP is the main factor for the Cas9sg transportation.

Subsequently, the cytosolic delivery is investigated by monitoring the co-localization of Cas9sg with the endosomal markers (early endosomal marker Rab5 and late endosomal marker Rab7) and the lysosomal marker LAMP1. As shown in Figure 5, some co-localization of Cas9sg with Rab5 is observed and scarcely any co-localization is visible with LAMP1 at 4 h incubation. Moreover, up to 12 h, the co-localization of Cas9sg with Rab5 and Rab7 are visible while hardly any co-localization is observed between Cas9sg and LAMP1. It demonstrates that Cas9sg are transported to early endosome and further delivered to late endosome rather than lysosome, which implies that Cas9sg is released from endosome based on CAP-associated molecule mechanism.

### 3.5. The Molecule Mechanism of Facilitating Endocytosis and Cytosolic Release of Cas9sg

The mechanism of RONS-enhanced uptake of bioactive macromolecules on the cellular and molecular levels is investigated. RONS generated from CAP exposure can affect the membrane permeability by lipid oxidation [36] and membrane leakage [37], and then enhance cellular uptake through passive diffusion [38]. Therefore, to investigate whether the endocytosis-independent pathways involved in the uptake of Cas9sg-488 are related to CAP-induced membrane leakage for passive diffusion, the cell membrane potential and permeability changes are evaluated by staining MCF-7 cells with Rhodamine123 (Rh123) and fluorescein diacetate (FDA) probes, respectively [39]. As shown in Figure 6a, the CAP-treated cells show less intense red fluorescence from Rh123 than the untreated cells and CAP-treated cells in the presence of the RONS scavenger NAC, confirming that membrane breakage associated with the decreased membrane potential after the CAP treatment arise from the increased RONS during CAP exposure. Moreover, after culturing for 6 h, the cells show increased red fluorescence of Rh123 (Figure 6b), suggesting that the membrane integrity and potential recover gradually to the same levels as the untreated cells. The results indicate that CAP-induced membrane leakage is temporary, and the cells can recover gradually after consumption of RONS. The membrane permeability is measured with the FDA probes. In comparison with the untreated cells and CAP-treated cells in the presence of NAC, green fluorescence from FDA in MCF-7 cells decreases upon CAP exposure and recovers to the same level as the untreated cells after incubation for 6 h (Figure 6c,d). It demonstrates that the changes of membrane integrity and potential induced by RONS during the CAP treatment can directly increase the membrane permeability, further facilitating the membrane transport of foreign species, such as Cas9sg. Therefore, RONS generated from CAP increases the membrane permeability and further results in passive diffusion of Cas9sg as an endocytosis-independent pathway.

The RONS-enhanced permeability is known to be beneficial to the transmembrane transport of other molecules besides the Cas9sg complexes. It has been reported that the CAP-enhanced membrane permeabilization accelerates ions to pass through the cell membrane, specially more influx of Ca^2+^ from extracellular media to cytoplasm via the voltage-dependent Cch1p channel [40]. To further investigate the molecule mechanism of CAP-mediated cell uptake and cytosolic release of Cas9sg, intracellular Ca^2+^ and ATP contents are measured with the Fluo 3-AM fluorescent probe and colorimetric assay, respectively. After incubation with Fluo 3-AM, the CAP-treated MCF-7 cells show robust green fluorescence compared to cells pre-treated with the RONS scavenger NAC and cells exposed to UV radiation alone (Figure 7a; Appendix A), confirming that the intracellular Ca^2+^ concentration depends on extracellular RONS generation by CAP. As an important cellular signaling molecule, intracellular Ca^2+^ is in volved in various cell metabolism regulation like ATPase activity [41]. Indeed, the intracellular ATP contents after CAP treatment decrease compared to the untreated cells (Appendix A), suggesting that the increased intracellular Ca^2+^ concentration activates intracellular ATPase. However, with the more influx of Ca^2+^, the ATP contents gradually increase at 2 h and 4 h incubation (Figure 7b), revealing that the high Ca^2+^ concentration leads to the accumulation of ATP for the inhibition of ATP hydrolysis. Interesting, ATP contents further increase in the presence of endocytosis inhibitors and at a low temperature, confirming that part of ATP contents are devoted to enhancing cellular uptake of Cas9sg via ATP-dependent endocytosis. Finally, up to 6 h incubation, the intracellular ATP contents decrease due to the consumption and exhaustion of Ca^2+^. Furthermore, Ca^2+^ and ATP are employed to enhance endosomal escape [42]. Therefore, we propose that the intracellular Ca^2+^ and ATP are also devoted to facilitating cytosolic release of Cas9sg from endosome, thus, decreasing Cas9sg delivery from endosome to lysosome, which contributed to protecting the bio-activity of Cas9sg for in vitro genome editing.

### 3.6. Genome Editing of Cas9sg after the CAP Treatment

To validate CAP as an efficient delivery approach for the CRISPR/Cas9 system, the A549/EGFP cells are employed to evaluate the gene disruption efficiency of Cas9sg upon CAP treatment. After the CAP treatment for different time, the A549/EGFP cells are incubated with 8 nmol L^−1^ Cas9sg (sgRNA targeting EGFP coding region), and cultured for five days at 37 °C. As shown in Figure 8a, green fluorescence of EGFP decreases gradually with the increase of CAP treatment time, suggesting that a longer CAP exposure time leads to more cytosolic delivery of active Cas9sg, and more efficient EGFP disruption. Flow cytometry is performed to quantitatively determine EGFP disruption, and 20.3% disruption efficiency in cells treated with CAP for 80 s is revealed (Figure 8b). Genome editing is explored in the HEK293T normal cells and MCF-7 cancer cells using active Cas9sg to target a non-coding region in chromosome 9. The cells are treated with CAP for 80 s, incubated in the presence of 8 nmol L^−1^ Cas9sg, and cultured for three days prior to genome extraction and T7E1 assays. CAP-mediated CRISPR/Cas9 transport results in indel efficiencies of 21.7% in the HEK293T cells and 30.2% in the MCF-7 cells (Figure 8c) compared to 19.6% and 27.5% observed from the Lipofectamine transfected cells as positive controls. In comparison with the wide used carrier-dependent (Lipofectamine transfection) and carrier-free (electroporation) strategies with the obstacles of costly agents and high cell death (Appendix A), CAP facilitates passive diffusion and indirect endocytosis of bioactive Cas9sg into cells for efficient gene editing.

## 4. Conclusions

A carrier-free CRISPR/Cas9 cellular transport system utilizing CAP is designed for effective genome editing. After CAP exposure, the MCF-7 cells show 88.9% uptake efficiency and 65.9% nuclear import efficiency after incubation with Cas9sg with a size of about 15 nm. The CAP-generated RONS leads to the variations in the membrane potential and permeability, thus, facilitating Cas9sg transport with passive diffusion. The change in the membrane permeability also leads to influx of Ca^2+^ into the cells, increases the ATPase activity, and accelerates hydrolysis of ATP to increase cellular uptake of Cas9sg by ATP-dependent endocytosis, including caveolae-mediated and clathrin-mediated endocytosis. CAP-mediated Cas9sg transportation shows 20.3% EGFP disruption efficiency in the A549/EGFP cells, and 21.7% and 30.2% indel efficiency in the HEK293T and MCF-7 cells, respectively. In comparison with carrier-dependent delivery suffering from problems such as biosafety and complicated preparation, this simple and effective CAP-mediated transport approach is applicable to delivery of drugs and other bioactive macromolecules in therapies.

## Figures and Tables

**Figure 1 biology-10-01038-f001:**
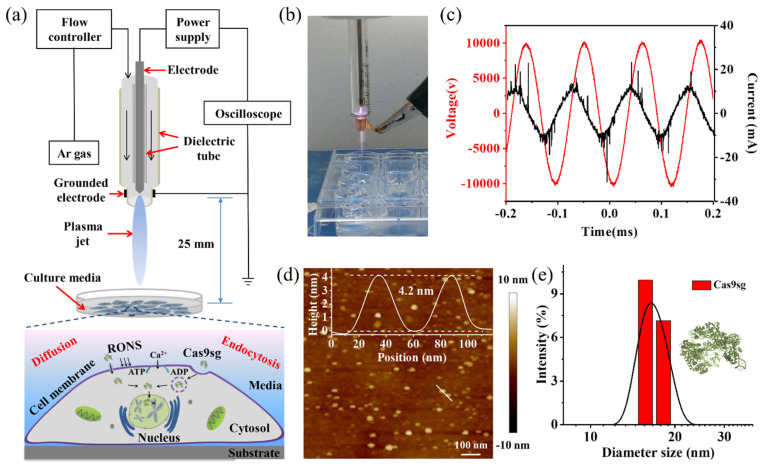
CAP-mediated cellular transport and characterization of Cas9sg: (**a**) Schematic illustration of CAP-mediated Cas9sg cellular transport for genome editing; (**b**) Image of CAP; (**c**) Voltage and current characteristic of CAP; (**d**) Morphology, and (**e**) Size distribution of Cas9sg determined by AFM and DLS measurements.

**Figure 2 biology-10-01038-f002:**
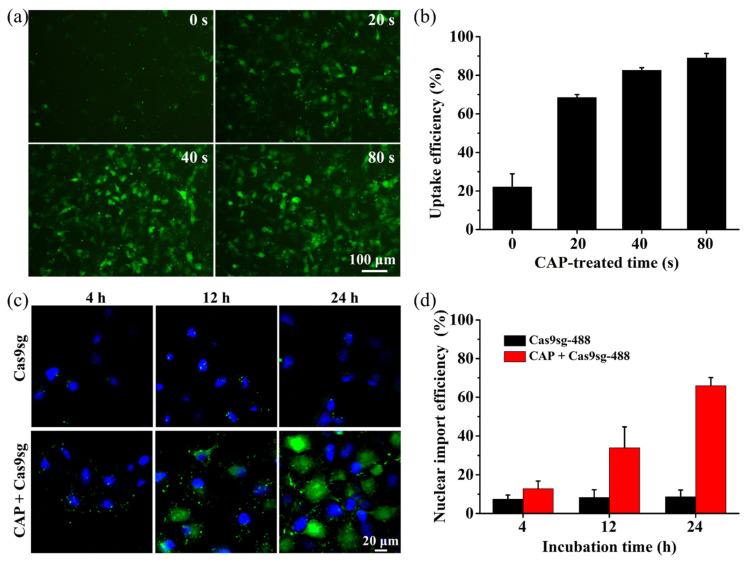
CAP-mediated cellular transport of Cas9sg: (**a**) Fluorescence imaging of the MCF-7 cells treated with Cas9sg-488 for 24 h after CAP exposure for various time periods; (**b**) Uptake efficiency of Cas9sg-488 in MCF-7 cells 24 h after CAP exposure for various time periods determined by flow cytometry; (**c**) Confocal fluorescence imaging and (**d**) Nuclear import efficiency of MCF-7 cells treated with Cas9sg-488 for different time intervals after CAP exposure for 80 s (C_Cas9sg-488_ = 8 nmol L^−1^). The data are shown as mean ± SD (*n* = 3).

**Figure 3 biology-10-01038-f003:**
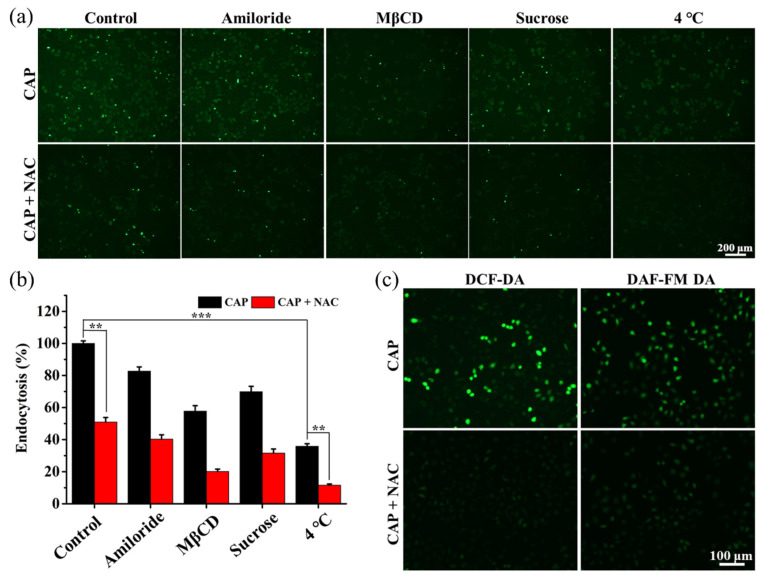
Mechanism of CAP-mediated cellular transport of Cas9sg: (**a**) Fluorescence images of Cas9sg-488 uptake by the MCF-7 cells pre-treated with different endocytosis inhibitors and at a low temperature (4 °C) upon CAP exposure for 80 s (C_Cas9sg-488_ = 8 nmol L^−1^); (**b**) Inhibition of endocytosis by different inhibitors and low temperature of 4 °C upon CAP exposure for 80 s; (**c**) Fluorescence images of MCF-7 cells treated with the DCFH-DA and DAF-FM DA probes for 2 h in the presence of the RONS scavenger NAC after the CAP treatment for 80 s. The data are shown as mean ± SD (*n* = 3). (**) *p* < 0.01, and (***) *p* < 0.001.

**Figure 4 biology-10-01038-f004:**
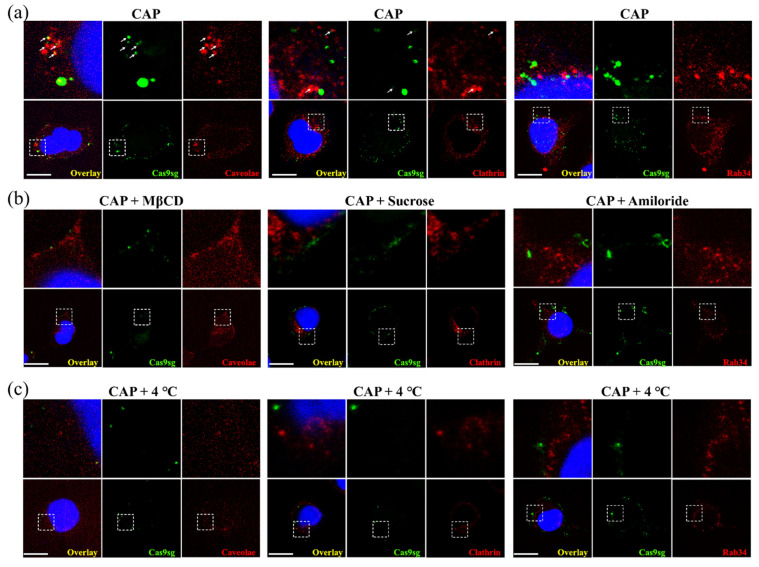
CAP-mediated cellular uptake of Cas9sg monitored by direct immunofluorescence: (**a**) Confocal images of CAP-treated cells incubated Cas9sg (green) for 4 h, and then the immunofluorescence experiments were performed to detect Caveolae, Clathrin-mediated endocytosis, and the macropinocytosis (red) with the primary antibody, respectively, against Caveolae, Clathrin, and Rab34; (**b**) Confocal immunofluorescence images of CAP-treated cells incubated Cas9 in presence of endocytosis inhibitors MβCD, Sucrose, and Amiloride, respectively; (**c**) Confocal immunofluorescence images of CAP-treated cells incubated Cas9 at a low temperature of 4 °C (C_Cas9sg_ = 8 nmol L^−1^).

**Figure 5 biology-10-01038-f005:**
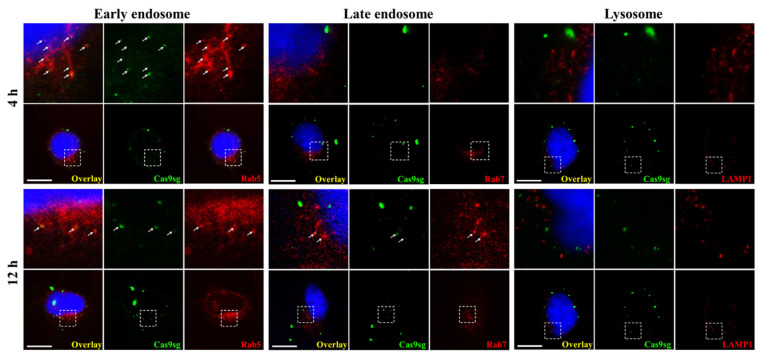
Confocal immunofluorescence images of co-localization of Cas9sg (green) with early endosomal marker Rab5 (red), late endosomal marker Rab7 (red) and lysosomal marker LAMP1 (red) (C_Cas9sg_ = 8 nmol L^−1^).

**Figure 6 biology-10-01038-f006:**
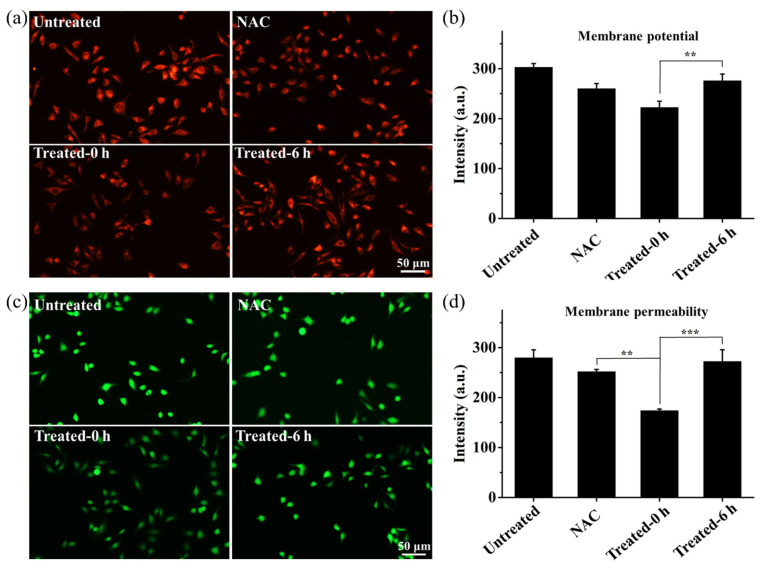
Determination of the membrane potential and permeability after the CAP treatment: (**a**) Fluorescence images and (**b**) Intensity changes observed from the MCF-7 cells stained with Rh123 upon CAP exposure of 80 s under different conditions to determine the membrane potential alteration; (**c**) Fluorescence images and (**d**) Intensity changes observed from the MCF-7 cells stained with FDA after the CAP treatment for 80 s under different conditions to evaluate of membrane permeability changes. The data are shown as mean ± SD (*n* = 3). (**) *p* < 0.01, and (***) *p* < 0.001.

**Figure 7 biology-10-01038-f007:**
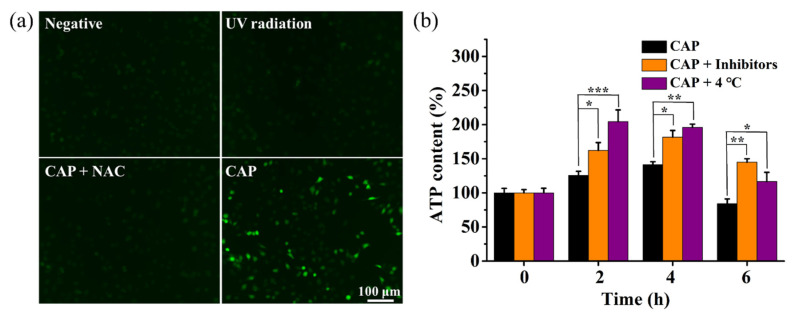
Determination of intracellular Ca^2+^ and ATP concentrations after the CAP treatment: (**a**) Fluorescence images of the MCF-7 cells treated with the Fluo 3-AM probe for 2 h under UV radiation and in the presence of RONS scavenger NAC after the CAP treatment for 80 s; (**b**) Time-dependent intracellular ATP content in the presence of endocytosis inhibitors and at low temperature of 4 °C after the CAP treatment for 80 s. The data are shown as mean ± SD (*n* = 3). (*) *p* < 0.05, (**) *p* < 0.01, and (***) *p* < 0.001.

**Figure 8 biology-10-01038-f008:**
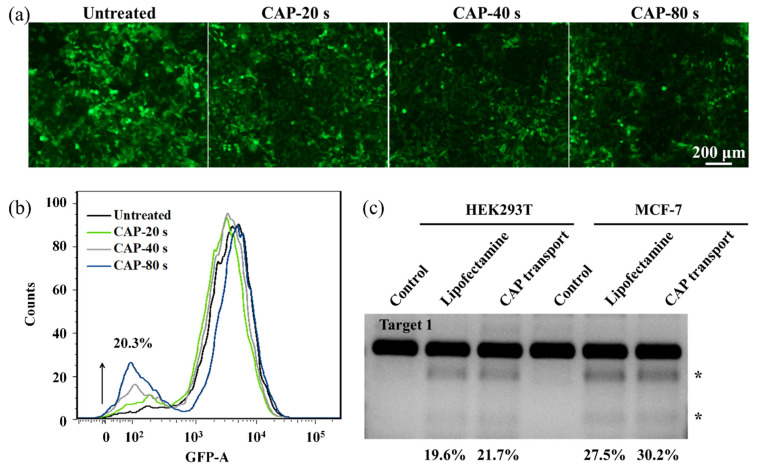
Genome editing of Cas9sg delivered by CAP treatment: (**a**) Fluorescence images of the A549/EGFP cells treated with Cas9sg for five days after CAP exposure for different time; (**b**) Flow cytometry of the A549/EGFP cells treated with Cas9sg for five days after CAP exposure for different time; (**c**) Indel analysis of HEK293T and MCF-7 cells treated with Cas9sg via Lipofatamine transfection and CAP treatment (C_Cas9sg_ = 8 nmol L^−1^). The asterisk (*) indicates the cleavage bonds of Cas9sg at Target 1 sites in chromosome 9.

## Data Availability

The data presented in this study will be made available upon request to the corresponding authors.

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
