# Peer review of "Carrier-Free Cellular Transport of CRISPR/Cas9 Ribonucleoprotein for Genome Editing by Cold Atmospheric Plasma"

_biology, 2021, doi:10.3390/biology10101038_

Round 1
Reviewer 1 Report
Report on the manuscript „carrier-free cellular transport of CRISPR/Cas9 ribonucleoprotein for genome editing by cold atmospheric plasma” by Cui et al. for publication in biology (MDPI).
The paper presents the results of a study aiming to investigate the non-thermal plasma effects (cold atmospheric plasma) of a self-constructed jet discharge on the Cas9 uptake into different cell lines and its intracellular behaviour.
In theory, this work is of interest for the community working on the understanding of the different mechanisms of action of non-thermal plasma in cell-cultural science, especially in the field of plasma medicine. Further, a possible carrier-free transport method concerning the CRISPR/Cas9 system is presented.
The whole article is written in a well prepared English and suits the formal standards.
Major/ general comments:
- Much minor spelling and grammar mistakes have been found. Please, check the article seriously.
In its current content, the manuscript can be published in biology (MDPI). However, a minor revision concerning the English language is recommended.
Reviewer 2 Report
Dear authors,
The manuscript biology-1378506-peer-review-v1, entitled ‘Carrier-free Cellular Transport of CRISPR/Cas9 Ribonucleoprotein for Genome Editing by Cold Atmospheric Plasma’ presents a cold atmospheric plasma (CAP)-based CRISPR/Cas9 carrier-free delivery platform and the corresponding mechanism related to efficient transportation has been studied. The authors found that the efficient production of bioactive species in culture media after plasma treatment alters cell membrane potential and permeability, which facilitates cytosolic delivery of active CRISPR/Cas9 components via passive diffusion and ATP-dependent endocytosis pathways, resulting in efficient genome editing and gene silencing. Moreover, this carrier-free strategy using CAP-based transportation may also be extended to other active biomolecules in drug delivery and gene therapy.
Characterization methods used by the authors are as follows: atomic force microscopy (AFM), zeta potentials and dynamic size, UV−vis absorption and fluorescence spectroscopy. Cell culture, cellular viability: MTT assays, in vitro fluorescence imaging, lipofectamine CRISPRMAX Cas9 transfection, flow cytometry of CAP-mediated cellular transport, CAP-mediated cellular transport, as well as intra and extracellular RONS levels, immunofluorescence studies, measurement of the intracellular ATP contents and indel analysis were used by the authors in their study. These biochemistry/biology study test are well presented in the manuscript.
However, the Plasma system is poorly represented, on page 2 lines 106-112. More details on the CAP properties ( U-I waveform, power, charge, energy, current, RONS, emitted spectra, streamer/wave, etc) and other useful infos upon the plasma source that helps / sustains the overall statements in the manuscript, starting from the title till conclusions.
I would suggest the authors to improve this paragraph (Plasma jet system) on page 3 with power supply manufacturer and characteristics, gas flow controller, and the device itself as dimension, materials, and gas purity.
Also, on page 7, the first Results paragraph, related to CAP. I would suggest the authors to introduce at least some electrical and optical characteristics of the discharge (voltage-current waveform, OES spectra, discharge photo?) to support the schematics (fig 1).
The overall manuscript text is written well, with good explanation and descriptions, except those mentioned above.
The conclusions are clearly presented and supported by the experimental results.
I propose this manuscript should be considered for publication in BIOLOGY Journal after meeting these suggestions.
Recommendation: MAJOR Revision.
Reviewer 3 Report
This is an interesting work and is high-impact to the plasma medicine community. The alteration of cell membrane permeability is a well-known effect of the plasma treatment, and this work gives a strong evidence. I cannot find any technical issues of the experimental design, results, and conclusions. I recommend an acceptance of this manuscript.
Reviewer 4 Report
The proposed work presents some interesting results on a carrier-free technique based on CAP for the delivery of CRISPR/Cas9 ribonucleoprotein. The paper is well presented and the scientific method is sound. There are nevertheless some important pieces of information missing from the Material and Methods section that must be introduced before the manuscript could be considered for publication. The Authors should take the following points into account in revising the manuscript:
- Line 67 : Not al CAP devices are based on a “jet” configuration. The sentence should be revised.
- M&M: Many important details on the CAP device are missing. Diameters of the electrode and quartz dielectric barrier for example. What is the purity of the Ar gas used? Is the voltage reported for the power supply peak-to-peak?
- M&M: It is not clear how the CAP treatment was performed. Were all the treatments performed on cells in 24-well plates? Were the cells covered by any liquid during the treatment? In which amount? What was the distance between the CAP device and the liquid? Was the sample at floating potential or grounded? Was the treatment static or was the CAP device moved?
- Introduction: There is very little reference to previous works on the transport of DNA into cells by means of CAP. The Authors should better highlight the novelty of their work with respect this piece of literature.
Round 2
Reviewer 2 Report
Dear Authors,
The revised version looks better than the first version of the manuscript. Needed information was added, both in the text and images part. However I would expect also some crucial information about the excited species in the plasma, so optical emission spectroscopy, at least at the level of board spectrum, e.g. visible or uv-visible range. In this manner, the information already presented in this revised version would be more of interest to the reader, and , of course, can support further explanations of the authors upon plasma effects.
With these info,s added, I believe that the manuscript will be suitable for publication in the Biology journal. So MINOR REVISION still needed.
